# Direct observation of long-lived cyanide anions in superexcited states

Xiao-Fei Gao[1], Jing-Chen Xie[1], Hao Li[1], Xin Meng[1], Yong Wu [2] & Shan Xi Tian [1✉]

The cyanide anion ($CN^-$) has been identified in cometary coma, interstellar medium, planetary atmosphere and circumstellar envelopes, but its origin and abundance are still disputed. An isolated $CN^-$ is stabilized in the vibrational states up to $\nu = 17$ of the electronic ground-state $^1\Sigma^+$, but it is not thought to survive in the electronic or vibrational states above the electron autodetachment threshold, namely, in superexcited states. Here we report the direct observation of long-lived $CN^-$ yields of the dissociative electron attachment to cyanogen bromide (BrCN), and confirm that some of the $CN^-$ yields are distributed in the superexcited vibrational states $\nu \geq 18$ ($^1\Sigma^+$) or the superexcited electronic states $^3\Sigma^+$ and $^3\Pi$. The triplet state can be accessed directly in the impulsive dissociation of $BrCN^-$ or by an intersystem transition from the superexcited vibrational states of $CN^-$. The exceptional stability of $CN^-$ in the superexcited states profoundly influences its abundance and is potentially related to the production of other compounds in interstellar space.

[1] Hefei National Laboratory for Physical Sciences at the Microscale, Collaborative Innovation Center of Chemistry for Energy Materials (iChEM), Department of Chemical Physics, University of Science and Technology of China, Hefei, China. [2] Institute of Applied Physics and Computational Mathematics, Beijing, China. ✉email: sxtian@ustc.edu.cn

Molecular anions have been discovered in dark clouds, prestellar cores, protostellar and circumstellar envelopes, Titan's atmosphere, and cometary comas, and they are involved in the substance evolutions of interstellar medium[1,2]. Although the detection of new molecular anions in space stagnates recently, laboratorial efforts are continuously being put into exploring anionic properties and identifying potential targets for astronomical observations[3]. Radiative electron attachment (REA), dissociative electron attachment (DEA), and ion-pair or polar dissociation are recognized as the typical pathways to produce anionic species, while a rapid anion–neutral reaction usually leads to the secondary anion[1–5]. As one of the smallest diatomic anions with the astronomical interest, cyanide anion ($CN^-$), was proposed to exist in Titan's ionosphere[6], subsequently detected at an altitude of 1015 km[7] and discovered in the carbon-rich star envelope IRC + 10216 (ref. [8]). Previously, its presence in the coma of comets was also mentioned[2,9]. Its productions by the REA to CN radical, the DEA to cyanopolyynes $HC_nN$ ($n = 1–3$) or MgCN/MgNC, and the $H^- + HC_nN$ reactions were evaluated with modeling calculations[1,4–10]. However, the role of REA process, initially proposed as the dominant mechanism[4], is still open to the debate[8–13], in particular, the $CN^-$ abundance in IRC + 10216 predicted with statistics theory calculations indicated a significant deviation from the observation[8,13]. Recently, the contribution from the DEA to $H_2CN$ in IRC + 10216 was eliminated either[14].

$CN^-$ at the electronically ground state $X^1\Sigma^+$ is known to be a stable anion because of the large electron affinity (EA = 3.862 ± 0.004 eV) of the neutral[15]. Furthermore, the $CN^-(X^1\Sigma^+)$ can be stabilized in various vibrational states ($v$) up to $v = 17$, while that in the higher $v$-state is believed to quickly decay via vibration-induced electron detachment[16]. Therefore, the vibrationally or electronically excited states above the electron autodetachment threshold of $CN^-$, namely, those locate 3.862 eV (EA) higher than $X^1\Sigma^+$ ($v = 0$), are assumed to have no contributions to the $CN^-$ abundance and some chemical reactions in the interstellar space[1,4–13]. However, a long-lived anion in high-lying excited states can be observed, if its vibration-induced electron detachment proceeds in a non-negligible time, for instance, on a time-scale comparable to the lifetimes (such as the nanoseconds to microseconds of $LiH^-$ and $OH^-$, ref. [16]). By the same token, the $CN^-$ could be long-lived in the vibrational states $v \geq 18$ or electronically excited states above the electron autodetachment threshold, but such a conjecture is subject to experimental validation.

Seeking the long-lived anion in the high-lying states is continued with great enthusiasm until now. The long-lived (milliseconds) $NC_4N^{-*}$ (ref. [17]) and (microseconds to seconds) $SF_6^{-*}$ (refs. [18,19]) were found to be populated in the vibrational states above their respective electron autodetachment thresholds. A high-lying spin-state $^6\Pi$ of $CO^-$, as a metastable anion, was predicted[20], which potentially elucidated the observation of long-lived ($10^{-5}$ s) $CO^-$ detected in the mass spectrometry experiment[21]. As illustrated in Fig. 1, various resonant states ($M^{*-}$) of a polyatomic anion can be formed in electron attachment, and they maintain the equilibrium structure of the neutral and locate energetically above the neutral state M[22]. The resonant states ($M^{*-}$) with various lifetimes (denoted with different energy widths for shape or Feshbach resonant states) decay quickly through dissociation or electron autodetachment[22]. Even if the bound states $M^{-*}$ and $M^{-**}$ exist in theory, the transformation from ($M^{*-}$) to $M^{-*}/M^{-**}$ is hardly accomplished by the structural relaxation in dozens of picoseconds, due to its much faster competitive processes such as dissociation and electron autodetachment of ($M^{*-}$). This is the primary reason that $M^{-*}$ or $M^{-**}$ is scarcely observed in the electron attachment

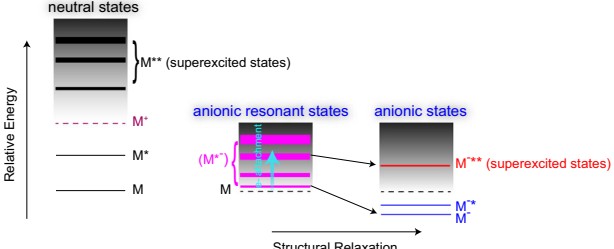

**Fig. 1 Comparison between neutral and anionic states.** Left: M, M*, M+, and M** represent the ground, excited, ionized, and superexcited states of a neutral molecule, respectively; Middle: ($M^{*-}$) at anionic resonant state is formed by electron attachment to M; Right: Stable $M^-$ and $M^{-*}$ can be formed by the structural relaxation from ($M^{*-}$), while a superexcited-state $M^{-**}$ (lying above M) is scarcely observed in experiments. Different widths of the energy levels of $M^{**}$, ($M^{*-}$), and $M^{-**}$ correspond to their lifetimes, due to the couplings with the electron continuum background (in black color).

experiment. Here superscripts $-**$ (or $-*$) and $*-$ represent the structurally relaxed (but in an excited state) and the resonant-state (within the neutral structure) anions, respectively; while $M^-$ is the anionic ground state. With reference to the neutral superexcited state $M^{**}$ which lies energetically above the cationic ground state $M^+$, $M^{-**}$, as an anionic state above the ground state M, is named as the anionic superexcited state. A comprehensive theoretical study of the ($CN^{*-}$) resonant states was reported by Harrison and Tennyson[23], indicating an energy sequence of resonant states $^3\Pi < {}^3\Sigma^+ < {}^3\Sigma^-$; while the $CN^{-**}$ superexcited bound states, the singlet $\Sigma$-, $\Pi$-, and $\Delta$-symmetric bound states[24], were predicted to be in an energy order of $^3\Sigma^+ < {}^3\Pi < {}^1\Pi$ (ref. [25]). Meanwhile, the equilibrium bond lengths of $CN^{-**}$ anions were close to that of the ground-state $CN^-$ or $CN$[24,25]. Therefore, once ($CN^{*-}$) is produced, its structural relaxation in the transformation ($CN^{*-}$) → $CN^{-**}$ would be considerably fast; alternatively, the $CN^{-**}$ superexcited bound states could be directly accessed by the electron attachment to CN radical. These two cases exhibit some possibilities to observe metastable $CN^{-**}$ in experiments. Here we show experimental evidence of $CN^{-**}$ produced in the DEA to cyanogen bromide (BrCN) and emphasize its profound roles in astrochemistry.

## Results and discussion

In this work, we report an experimental evidence that the $CN^{-**}$ species are produced in the DEA to BrCN,

$$e^- + BrCN \rightarrow Br(P_{3/2})/Br^*(P_{1/2}) + CN^-/CN^{-**} \quad (1)$$

where the $CN^-$ yield is populated in the vibrational states ($v \leq 17$) at the low electron attachment energy ($E_e$), while the $CN^{-**}$ produced at the high $E_e$ value is in the vibrationally or electronically superexcited states. In the previous studies, the $CN^{-**}$ species cannot be identified in the $CN^-$ production efficiency curve[26] and the low vibrational-state ($v < 9$) $CN^-$ yields were found in the lower attachment energy range ($E_e = 1.07–1.97$ eV)[27]. Using the high-resolution time-sliced velocity map imaging (VMI) apparatus[27–30], which was developed on the basis of our previous one[31,32], we record the velocity images of the $CN^-/CN^{-**}$ yields in the higher $E_e$ range from 3.57 to 6.57 eV, and the results are shown in Fig. 2.

According to the energy conservation, the internal energy ($E_{int}$) of $CN^-$ is determined with,

$$E_{int}(CN^-) = E_e - E_{th} - E_k \quad (2)$$

where $E_{th}$ is the DEA threshold [−0.13 eV for Br + $CN^-(X^1\Sigma^+$,

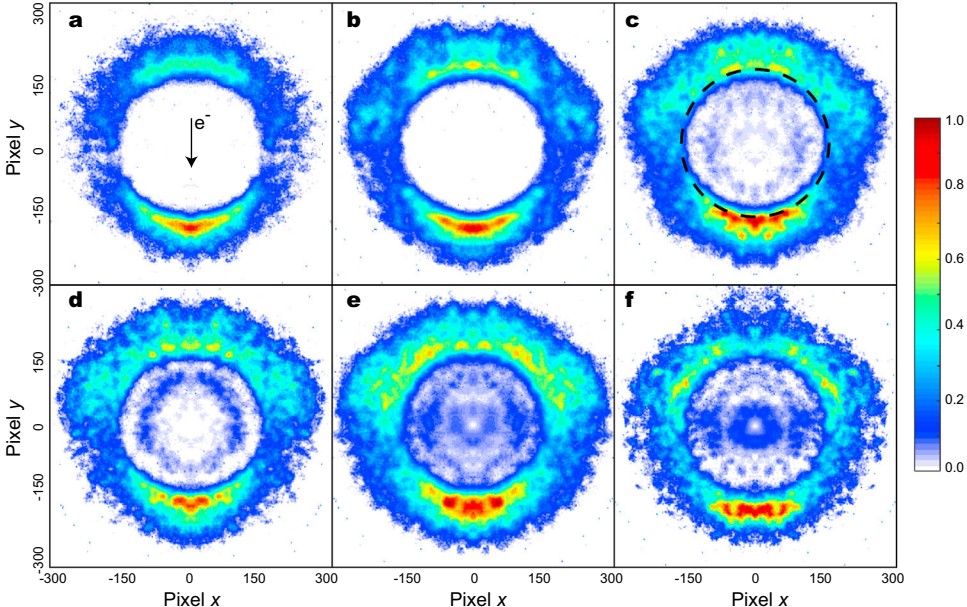

**Fig. 2 Time-sliced velocity images of CN$^-$.** These CN$^-$ ions are produced in dissociative electron attachments to BrCN at the attachment energies of 3.57 (**a**), 4.57 (**b**), 5.07 (**c**), 5.27 (**d**), 6.07 (**e**), and 6.57 (**f**) eV. **a** The electron incident direction is denoted with an arrow in the middle of image, which defines the forward- and backward-scattered distributions of CN$^-$ yields. **c** The internal energy of the CN$^{-**}$ yields distributed in the inside region of a broken circle embedded on the image is higher than the electron autodetachment threshold. These long-lived metastable CN$^{-**}$ ions are also observed in (**d-f**). The anionic intensity in each sliced image is normalized with the weighing factors of different-sized Newton spheres, and each image is plotted independently in a relative intensity scale.

$v = 0$); 0.33 eV for Br* + CN$^-(X^1\Sigma^+, v = 0$)] and $E_k$ is the total kinetic energy release of the DEA products. In a velocity image, the anion having a small kinetic energy locates a position close to the image center, and vice versa. To our surprise, besides the outside strong signals of the fast CN$^-$, some slow anions emerge in the inside region at $E_e = 5.07$ eV and their intensities are gradually enhanced as $E_e$ increases. According to Eq. (2), these slow CN$^-$ must reserve the higher $E_{int}$. For example, a broken circle embedded on the image of Fig. 2c denotes the velocity position of the CN$^-(X^1\Sigma^+)$ at the vibrational state $v = 17$, thus the inside signals demonstrate the existence of the superexcited-state CN$^{-**}$ yields. As mentioned above, these superexcited states should be CN$^-(X^1\Sigma^+)$ in the vibrational states $v \geq 18$ or that in the high-lying electronic states[24,25].

On the other hand, the velocity distributions of the inside anions diversify remarkably with the $E_e$ increase. From Fig. 2c to f, the inner ring-like distribution becomes most distinct at $E_e = 5.27$ eV and it is weakened subsequently; an additional smaller ring appears around the image center at 6.07 eV and turns to be prominent at 6.57 eV. Besides the forward-backward scattered distributions of the outside anions, the anions with the inner ring-like distribution show the relatively strong intensities in the sideward and backward directions (Fig. 2d), then their intensities in the sideward are slightly enhanced (Fig. 2e, f). Although the potential energy surfaces of BrCN$^-$ in the low-lying states have been successfully evaluated in our recent study[27], those in the high-lying states are still unavailable to date. Moreover, an explicit theory to describe the nuclear-electron motion couplings is required to understand the above variances of the CN$^-$ angular distributions. Therefore, we will focus only on the $E_{int}$ distributions of the CN$^-$ yields and identify their internal states by the thermodynamics analyses.

Figure 3 exhibits the CN$^-$ kinetic energy distributions, in which each point (solid circle) is obtained by summarizing the ion intensity for a common velocity within whole scattering angles. In the present $E_e$ range, the dissociation channels leading

to CN$^-$ and Br or Br* are accessible. According to Eq. (2) and the momentum conservation, different CN$^-$ quantum states for the channels to CN$^-$ + Br and CN$^-$ + Br* are assigned in Supplementary Fig. 1 of Supplementary Note 1, while only the former channel is presented in Fig. 3 for the purpose of clear visibility. As shown in Fig. 3a, b, the vibrational states $v \leq 17$ of the ground-state CN$^-(^1\Sigma^+)$ are assigned and the states of $v = 9$ (at $E_e = 3.57$ eV) and 13 (at $E_e = 4.57$ eV) are responsible for the maxima of the profiles.

At the higher $E_e$ values, some fine structures are observed at the left sides of Fig. 3c–f. They, together with a left part of the big band, are attributed to the superexcited-state CN$^{-**}$ yields (shaded in yellow). Considering the contributions of $^3\Sigma^+$ and $^3\Pi$ states, we reproduced the potential energy curves of these two and $^1\Pi$ superexcited states on the basis of our high-level calculations. The present results are shown in Fig. 4, in agreement with the previous ones[25]. Using the potential energy curves, we further derived the vibrational energy levels of $^3\Sigma^+$, $^3\Pi$, and $^1\Pi$ states (more details can be found in Supplementary Tables 1 and 2 of Supplementary Note 3). As depicted in Fig. 3c, d, the low vibrational states $v = 1$ or 2 of $^3\Sigma^+$ should be responsible for the small peak around the kinetic energy of 0.40 eV. At $E_e = 6.07$ and 6.57 eV, two small peaks at 0.05 and 0.40 eV are attributed to $v = 7, 10$ (Fig. 3e) and 9/10, 13 (Fig. 3f) of $^3\Sigma^+$, respectively. Meanwhile, the contributions from the specific vibrational states of $^3\Pi$ are possible. Note that the highest intensity (shaded in yellow) of the big band in Fig. 3e (or 3f) may be owed to the superexcited vibrational states $18 \leq v \leq 23$ (or 26) of $^1\Sigma^+$, the lower vibrational states of $^3\Sigma^+$, or both of them. Here we conclude that the superexcited vibrational states are highly preferable because their corresponding part is a portion of this big band, distinctly different from the isolated small peaks observed on the left side. Furthermore, the CN$^{-**}$ yields in $^3\Sigma^+$ and $^3\Pi$ states are expected to be directly produced in an impulsive DEA process, leading to the small peaks at 0.05 and 0.40 eV and their anisotropic distributions (see Fig. 3c–f). On the other hand, as an example, the

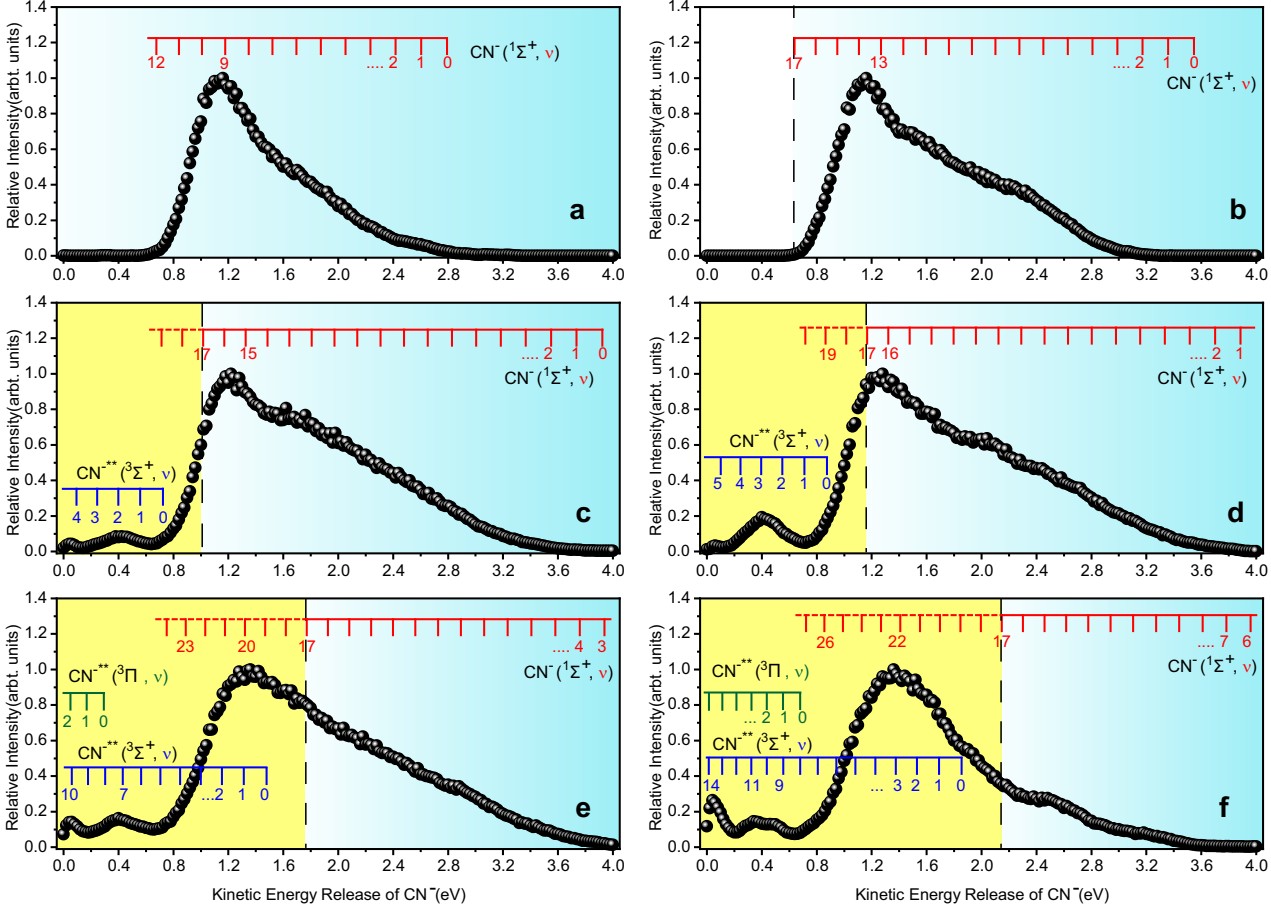

**Fig. 3 Kinetic energy distributions of CN⁻.** These CN⁻ ions are produced in dissociative electron attachments to BrCN at 3.57 (**a**), 4.57 (**b**), 5.07 (**c**), 5.27 (**d**), 6.07 (**e**), and 6.57 (**f**) eV. The vibrational states of the ground ($^1\Sigma^+$) and superexcited ($^3\Sigma^+$, $^3\Pi$) states are assigned for $Br(^2P_{3/2}) + CN^-/CN^{-**}$, and two regions, the superexcited-state one (shaded in yellow) and the low vibrational state region of the ground-state $^1\Sigma^+$ (shaded in cyan), are divided with a vertical dashed line which lies at the left side of $^1\Sigma^+$ ($v = 17$) indicates the threshold of the electron autodetachment of CN⁻.

anisotropic angular distributions in Supplementary Fig. 2 of Supplementary Note 2 show the remarkable differences among the $CN^{-**}$ at $^3\Sigma^+/^3\Pi$ state, the $CN^{-**}$ at the superexcited vibrational states ($v \geq 18$) of $^1\Sigma^+$ and the CN⁻ at $v < 17$ of $^1\Sigma^+$. It is beyond the present scope to gain more details about their different DEA dynamics, and the sophisticated theoretical calculations are demanded.

Besides the direct production of the $^3\Sigma^+$- / $^3\Pi$-state $CN^{-**}$ (its branching ratios are shown in Supplementary Fig. 3 of Supplementary Note 4), an intersystem transition from the superexcited vibrational state of $^1\Sigma^+$ to the bound state $^3\Sigma^+$ or $^3\Pi$ is feasible. Similarly, an inverse internal conversion from a vibrationally-hot ground electronic state to a bound electronic excited state was observed[33]; more recently, thermionic emission on a millisecond timescale from the vibrationally-hot anion was also reported[34]. As illustrated in the inset panel of Fig. 4, the intersystem transition (denoted as pathway **2**, while pathway **1** represents a direct pumping to $^3\Sigma^+$) from the vibrational states above $v \geq 20$ of $X^1\Sigma^+$ to $^3\Sigma^+$ state is conceptually analogical to the singlet-triplet state intersystem transition of the neutral species. Once the triplet states $^3\Sigma^+$ or $^3\Pi$ is populated (regardless of pathway **1** or **2**), the fluorescence decay of $CN^{-**}$ is unpermitted due to the spin-forbidden rule. We tried to detect the possible phosphorescence of $CN^{-**}(^3\Sigma^+$ or $^3\Pi) \rightarrow CN^-(X\,^1\Sigma^+)$ or fluorescence of $CN^{-**}$ (high-lying singlet states[25]) $\rightarrow CN^-(X\,^1\Sigma^+)$ using the spectrometer combined with the present apparatus[35] but no

photoemissions were detected, which indicates an ultralong lifetime of the triplet-state $CN^{-**}$ and the absence of the superexcited singlet-state $CN^{-**}$. Considering the flying time in the VMI measurements (see Supplementary Fig. 4 and Supplementary Note 5), we conclude that the lifetime of the superexcited triplet-state $CN^{-**}$ is more than 5 µs.

The lifetime of the resonant-state anion is influenced not only by the electron autodetachment but also the nuclear motions. With the C–N bond elongation (from the neutral equilibrium bond length 1.17 to 1.37 Å), the R-matrix calculation predicted a sharp increase of the lifetime (approximately as $\hbar/2\Gamma$, where $\Gamma$ is the energy width) of the resonant state $^3\Sigma^+$ (ref. [23]), indicating again that the structural relaxation does enhance the anionic stability. Despite the lack of a theoretical model of the vibration effect on resonant-state lifetime, the anionic lifetime is potentially enhanced by the molecular vibrations[17–19] or rotations[36,37]. A high vibrational-state density[17–19] is expected to prolong the lifetime[38], while the highly rotating anion at the electronic ground state can survive further because an energy barrier impedes the dissociation[37]. Here some rotational states of the $CN^-/CN^{-**}$ yield may be populated, corresponding to the isotropic background of the outside anionic signals in Fig. 2c–f. Besides the possible roles mentioned above[37,38], the long lifetime of the ro-vibrationally superexcited $CN^{-**}$ facilitates, in turn, the intersystem transition $^1\Sigma^+$ ($v \geq 20$) $\rightarrow ^3\Sigma^+$. More importantly, spin-orbit coupling in this anionic-state transition could be further

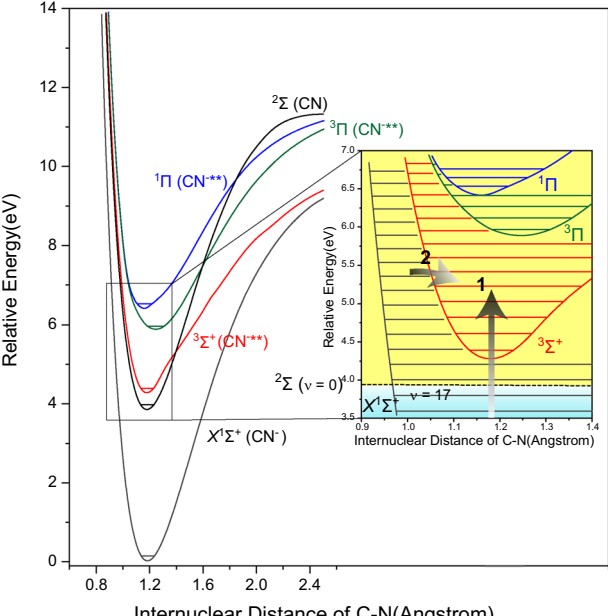

**Fig. 4 Potential energy curves of CN($X^2\Sigma$), CN⁻($X^1\Sigma^+$) and CN⁻**($^3\Sigma^+$, $^3\Pi$ and $^1\Pi$).** In the inset panel, the horizontal broken line denotes the zero-point vibrational energy level of CN ($X^2\Sigma$), the vibrational (horizontal parallel lines) and electronic states above these broken lines are the superexcited states and embedded on the yellow background, **1** represents the direct production of CN⁻** ($^3\Sigma^+$, $^3\Pi$) in the DEA process of BrCN or by electron attachment to CN($X^2\Sigma$) and **2** is the intersystem transition from the high-lying vibrational states of CN⁻($X^1\Sigma^+$) to the superexcited state $^3\Sigma^+$ of CN⁻**.

strengthened with help of the electron continuum background, thus this intersystem transition could be achieved more efficiently than that of the neutral.

Here we report an experimental evidence that the CN⁻** anions in the superexcited electronic states $^3\Sigma^+$ and $^3\Pi$ and the vibrational states $v \geq 18$ of the ground state $X^1\Sigma^+$ are produced in the DEA to BrCN, and propose that the intersystem transition $X^1\Sigma^+(v \geq 20) \rightarrow {}^3\Sigma^+$ of the free fragment CN⁻** is favorable in dynamics. According to the operation condition of our VMI measurements, the CN⁻** lifetime is expected to be not less than 5 µs. This superexcited-state species is unlikely produced in the DEA to NCCN[39] or other polyatomic molecules (such as CH₃CN[40]), since the excess energy is apt to be transformed into the internal or translational energies of the multiple neutral products. The energies of the thermally equilibrium electrons in astrophysical environments are much less than 1 eV, but there are a lot of low-energy electrons, in particular, a peak distribution exhibiting an appealing coverage of 4–6 eV, at the altitude of 1015 km in Titan's ionosphere[7]. The free electrons with above energies can be also produced promptly in the irradiation ionizations of the interstellar substances. Therefore, as described in Fig. 4, the superexcited-state CN⁻** are possibly produced by the electron attachment to CN radical (via pathways **1** and **2**), because of many free CN radicals in the interstellar space. Furthermore, the long-term arguments[4,10–13] about the CN⁻ abundance in IRC + 10,216 are hopefully settled down if the contribution of the superexcited-state CN⁻** is considered. In addition, a CN⁻**-involved reaction proceeds much more readily than that of the ground-state CN⁻, for example, NCO⁻ in the L134N dark molecular cloud[41,42] could be formed in a barrierless reaction CN⁻** + $O_2 \rightarrow$ NCO⁻ + O. The present finding arouses attention on the potential roles of the superexcited bound states of anionic species in astrochemistry.

## Methods

**Experiments.** In our laboratory, anionic high-resolution velocity (or momentum) imaging has been realized[27–30], owing to the application of a trochoidal electron monochromator. In the experiments, within a crossed-beam arrangement, the monochromatized pulsed electrons (with an energy spread about 100 meV here, along y axis) are guided to the reaction region with a homogeneous magnetic field (70 G) which is produced with a pair of Helmholtz coils; then the anionic yields are pushed out, accelerated, and flying (along z axis) through the VMI lens system. In the flight of a given type of anions, different kinetic energies correspond to a set of concentric Newton spheres with different radii. The accelerated anions are detected with double multichannel plates plus a phosphor screen (in x–y plane). The central time-sliced image of the Newton spheres is recorded with a CCD camera mounted behind the phosphor screen and by applying a short high-voltage pulse on the rear multichannel plate. This pulse is also used as the mass gate to selectively detect the anionic yields. The solid sample BrCN is purified with several liquid-nitrogen freeze-pump-thawed cycles before the measurements, then introduced into the chamber with an inlet nozzle. Its volatility at the ice-water mixture temperature is high enough to create a sufficient concentration of target molecules in the gas phase, and the ambient pressure is controlled at ca. $10^{-6}$ Torr during the VMI measurements.

**Calculations.** The potential energy curve calculations are carried out for the neutral and anionic cyano by using the internally contracted multi-reference configuration interaction (icMRCI)[43] and the modified aug-cc-pVQZ basis set[44] by supplementing one diffuse s-function both on C and N atom. The active space consists of the valence orbitals $3\sigma$–$6\sigma$, $1\pi$ and $2\pi$, while the core orbitals, $1\sigma$ and $2\sigma$, are frozen. The MRCI treatment of electronic states is acquired in the state-averaged complete-active-space self-consistent field calculations on given spin-space symmetry species. More information can be found in Supplementary Note 3.

## Data availability

Vibrational states assignments (Supplementary Note 1), angular distribution of CN⁻/CN⁻** (Supplementary Note 2), vibrational state levels of $^3\Sigma^+$ and $^3\Pi$ states (Supplementary Note 3), branching ratios of the CN⁻** ($^3\Sigma^+/^3\Pi$) (Supplementary Note 4), determination of lifetime of CN⁻** (Supplementary Note 5) and the supplementary references are available as Supplementary Information in the online version of the paper. The data that support the present findings are available on request to the corresponding author.

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

## Acknowledgements

This work is supported by the Natural Science Foundation of China (Grant Nos. 21625301, 21727804) and the Ministry of Science and Technology of China (Grant No. 2017YFA0303502). Lei Xia and Xu-Dong Wang participated in the early work on this topic.

## Author contributions

S.X.T. supervised the study. X.-F.G., H.L., and X.M. carried out the experimental measurements. J.-C.X. performed the theoretical calculations with help from Y.W. and X-.F.G. J.-C.X. did the data analyses. All authors discussed the results and contributed to the manuscript.

## Competing interests

The authors declare no competing interests.
