## [Peer Review File · Communications Chemistry]

Reviewers' comments:

Reviewer #1 (Remarks to the Author):

See the attached file. 
Reviewer #2 (Remarks to the Author):

Please see review.pdf 
Reviewer #3 (Remarks to the Author):

This is an interesting paper that reports on some surprising results for the observation of superexcited states (that is, states, both vibrational and electronic, that are above the electron autodetachment energy) for CN^- . This has implications for the formation of small molecular anions in astrophysical environments, which the authors note to some extent. The work itself is worthy of publication in *Communications Chemistry*, in my view, but the English of the manuscript needs to be significantly improved. The biggest issue is the use of incorrect words in many, many places. For example, in the Introduction, the authors state that "...its existence in the coma of comet Halley was also of great concern." Clearly "concern" is not what the authors mean here. Unfortunately, there are many examples of this and in some cases the meaning is changed.

Some minor concerns: (a) The existence of CN^- in astrophysical environments was much earlier than stated as being in Titan's atmosphere. There are papers going back at least to the early 1980s, if not earlier. (b) In the case of larger molecular ions, such as NC_4N^- and SF_6^- , what role is played by redistributing the excess energy around the many vibrational degrees of freedom, many of which may not lead to autodetachment? (c) What electronic structure package was used to produce the potential energy curves in Fig. 3? MOLPRO? In fact, I think the authors should give more details of the calculations in the Supplemental material, enough so that Fig. 3 could be reproduced, which it could not be with the level of detail currently given.

Reviewer # 1

The paper describes an observation of the CN^- anion in a superexcited state. This means that the anion has a sufficient energy for electron autodetachment but is observed on a microsecond time scale. This result itself is fundamentally interesting – very few diatomic species with the total energy in continuum have been detected on such a long time scale. The findings thus should be published in a broad-audience journal such as *Comm. Chem.* However, in its present form, the manuscript has several important deficiencies.

Two major issues:

1. The main experimental result is that the measured CN^- kinetic energies are so low, that the internal energy of CN^- has to be higher than its electron affinity. In the discussion, the authors automatically assume that CN^- is either in its ground electronic and high vibrational state, or in its excited electronic state. That is actually only one of the possible interpretations. The authors almost completely neglect the possibility that the internal energy is stored in rotations. Only one sentence mentions rotations and that sentence is completely generic. (line 210). In fact, a large amount of energy can be stored in the rotational motion, which can directly lead to a long lifetime of the anions. The examples from the literature are hydrogen molecular anions (doi: 10.1103/PhysRevLett.94.223003), hydrogen halides (doi: 10.1016/j.ijms.2008.07.034) or metal dimer anions (doi: 10.1103/PhysRevLett.94.113201).

Reply: We appreciate this suggestion. Molecular rotation, as another type of nuclear motions, potentially influences the lifetime of anion. The references and more discussions about this issue are supplemented in the revised manuscript.

Related question: What is the symmetry of the involved resonant state? Is the dissociation of BrCN^- resonant state allowed in the linear geometry? If not, the dissociation process would need to proceed via out-of-line bending which could indeed create CN^- in high rotational states.

*Reply: Up to date, there are still no theoretical predictions about the resonant states of BrCN^- in the energy range investigated here. In our previous study (Gao X.-F., et al. *J. Phys. Chem. Lett.* 2020, 11, 9110), the potential energy surfaces of the low-lying states of BrCN^- were predicted and evaluated with the high resolution VMI measurements of the CN^- products. In that work, the CN^- yields in high rotational states were observed.*

2. What worries me a bit is that there is very little information about the experimental setup available. The authors report resolution of 100 meV, which is very good, considering that they operate in a pulsed mode. The paper says that the setup is described in refs. 25 and 26, but this is not exactly true. Ref. 25 contains another reference to an unpublished work (which I failed to find), ref 26 is a one-page conference abstract. At the same time, the good resolution is crucial in the author's conclusion (high internal energy content of CN^- is valid only in a certain range of measured fragment energies). The paper, or at least the SI, should specifically contain the following information:

- Does the relative CN^- yield as a function of electron energy agree with other published data? (which are cited deficiently, as I detail later). Ideally, show a comparison of the present ion yield with the earlier reports in SI.

*Reply: Ref. 26 (Xu-Dong Wang et al 2020 *J. Phys.: Conf. Ser.* 1412, 052010) is NOT a one-page conference abstract, which is a five-page progress report on an international conference. A*

comparison between our previous and present apparatuses was recently discussed in our review article (*Chinese Journal of Chemical Physics*. 2020, 33, 521-531, this journal can be found in AIP web page) which is now cited in the revised manuscript. As shown in the following figure, the CN^- production efficiency curve obtained in this work is compared with the previous report (Bunning et al., *JPC* 1996, 100, 19740).

Figure. Comparison of the CN^- production efficiency curves between ours (solid curve) and the previous report by Brunning et al. (Illenberger's group, solid circles).

The data (solid circles) reproduced from the literature (*JPC* 1996, 100, 19740) were renormalized for comparison, where two peaks around the electron attachment energies of 1.8 and 5.5 eV are also observed in this work. The profile and intensity deviations for the second band should be due to the lower collection efficiency of the fast ionic products by utilizing a small aperture in front of a quadrupole mass filter (*JPC* 1996, 100, 19740). This issue has been pointed out for several times in our published articles (for instance, *JPCA* 2019, 123, 9089–9095).

- How was the electron energy resolution determined? Ideally, show a spectrum demonstrating this resolution. - What was the pulse duration and did the authors observe any multiple collision effects such as in their ref. 25? - How was the kinetic energy of the fragments calibrated?

Reply: Details about above issues are beyond the scope of this work. As the reviewer knows, our experimental procedure, including the energy calibrations and the elimination of the interference from multiple collisions, was presented in the other articles (some of them were cited as the references in the manuscript). The time sequence of the pulses used in the experiments can be found as SI (section 5).

- What is the flight time of the CN^- ions?

Reply: The flight time of the CN^- ions is the key to estimate the life time of the superexcited-state CN^- . In section 5 of SI, we described how to derive the low limit of the life time of the superexcited-state CN^- .

- The authors do not discuss Br^- fragment, even though it is produced at the current energy range, and according to Brunning et al. (not cited!) even much stronger. Did the authors measure Br^- kinetic energy distribution? Brunning et al. measured the mean Br^- kinetic energy to be 0.55 eV,

this could serve as a comparison for the present data.

Reply: We are sorry for missing a reference to the report by Brunning and co-workers. Now it, together with the others, is cited in the revised manuscript. We did record the images of Br^- products, but they (unpublished) are not related to the present topic.

There are important bibliographic oversights. Consulting the following publications could bring more light on the problem.

- Brunning et al. (JPC 1996, 100, 19740) did an in-depth study of DEA to BrCN. How do the results compare?

- Parthasarathy et al. (JCP 2001, 114, 7962) measured kinetic energies of CN^- from BrCN following the electron transfer from Rydberg atoms and observed that almost all the available energy goes into the translational energy. Even though such charge transfer better corresponds to DEA at very low electron energies, this striking difference should be discussed.

- Royal and Orel (JCP 2006, 125, 214307) did theoretical analysis of resonances in BrCN. They provide information about the symmetries of resonances, asymptotic states and the dissociation dynamics.

- Nag et al. (PRA 2019, 99, 052705) did VMI study of CN^- production from NCCN. They observed a similar effect, very low translational energy of CN^- and concluded that it is left in high vibrational and rotational states.

Reply: JPC 1996 and PRA 2019 are cited and their results are discussed in the revised manuscript. JCP 2001 (Parthasarathy et al.) and JCP 2006 (Royal and Orel) focused on the DEA processes at the low attachment energies, thus they are not closely related to the present study.

Minor issue: The first sentence is ‘Lots of molecular anions have been discovered.’. Exactly seven different anions have been identified. Compared to hundreds of neutrals and tens of different cations, the use of ‘Lots’ is not exactly justified. The grammar is not ideal, however, it is premature to point out individual errors in this stage of the review process. Perhaps the most important one: in the title should be “in” instead of “at”.

Reply: We looked through the manuscript and some English errors or misleading sentences were corrected or revised.

Reviewer # 2

There is much to like about the science detailed in this paper, so I would like to see it published. However, there are several issues that I believe need to be addressed first.

1. It would help a lot if the authors could define clearly near the beginning of their paper exactly what they mean with the notations * and ** and *- and -*, etc. I think I followed it correctly, but it would be best to make these things clear from the beginning.

Reply: These notations have been given in the text and the caption of Fig. 1. A sentence describing the difference between $^-$ and $^-*$ is added in the revised manuscript.*

2. There are several minor English grammar and word-usage problems; I don't know how to address this, but it would be worth asking the authors to have a native English speaker go through the manuscript.

Reply: We carefully checked up and some English errors or misleading sentences were corrected or

revised.

3. This paper does give good evidence to support the claim that CN⁻ can be formed in high vibrational levels when an electron attaches to BrCN and dissociates it into Br + CN⁻. It also gives good evidence that at somewhat higher electron kinetic energies the CN⁻ can be produced in one or more excited electronic states. However, I don't see any evidence to support the claim made on line 227 that an electron can attach directly to the CN radical to form the so-called superexcited CN^{-**} species. In other words, I think the science described in this paper is solid in terms of showing some interesting process that happen with an electron strikes BrCN, but I don't see evidence to suggest that the findings provide evidence that CN⁻ anions can be formed by electrons directly attaching to CN radicals.

*Reply: Yes, there are still no direct evidences on the superexcited-state CN^{-**} produced in the electron attachment to CN, but the possibility cannot be simply ruled out. The present study should inspire the further investigation on this process, because there are lots of neutral CN radicals and free electrons in interstellar space or the planet's atmosphere. In the revised manuscript (Conclusion section), we also stress on the unfeasibility to produce CN^{-**} by the DEA to the large molecule containing the -CN group(s).*

The point #3 raised above makes me think that the information provided in this manuscript does not relate to how these CN^{-**} species might be formed in the astrophysical environments discussed here. I think it would be worth asking a person who knows more about astrophysics than I do to weigh in on this matter.

In summary, I suggest that the authors be asked to argue better how their findings might relate to how CN^{-**} might be formed in the environments they talk a lot about and I suggest that input from a person who knows more about astrophysics than I do be asked for input.

*Reply: Prior to this work, the possible existence of the superexcited-state CN^{-**} was never mentioned in astrophysics or astrochemistry, on the basis of the intuitive judgement about the stability of anionic species, namely, an anion with the internal energy higher than the electron detachment threshold (or the electron affinity of the neutral) is extremely short-lived. About 5 microsecond (or longer) of the lifetime of CN^{-**} found in this work breaks through this long period of neglect and prejudice.*

Reviewer # 3

This is an interesting paper that reports on some surprising results for the observation of superexcited states (that is, states, both vibrational and electronic, that are above the electron autodetachment energy) for CN⁻. This has implications for the formation of small molecular anions in astrophysical environments, which the authors note to some extent. The work itself is worthy of publication in Communications Chemistry, in my view, but the English of the manuscript needs to be significantly improved. The biggest issue is the use of incorrect words in many, many places. For example, in the Introduction, the authors state that "...its existence in the coma of comet Halley was also of great concern." Clearly "concern" is not what the authors mean here. Unfortunately, there are many examples of this and in some cases the meaning is changed.

Reply: We modified the manuscript and some errors or improper expressions are corrected.

Some minor concerns: (a) The existence of CN⁻ in astrophysical environments was much earlier than stated as being in Titan's atmosphere. There are papers going back at least to the early 1980s, if not earlier.

*Reply: The larger anions such as C_nN⁻ (n = 2, 3, ...) were noticed much early due to the interest in astrophysics and astrochemistry, however, CN⁻ as the smallest diatomic anion, rather than its neutral, was discovered very later. We also found a paper which mentioned the possible existence of CN⁻ in interstellar space (A. Wekhof, *The Moon and the Planets*, 1981, 24, 157-173) and cited it in the revised manuscript. We rewrote the sentence about the CN⁻ findings in the revised manuscript.*

(b) In the case of larger molecular ions, such as NC₄N⁻ and SF₆⁻, what role is played by redistributing the excess energy around the many vibrational degrees of freedom, many of which may not lead to autodetachment?

Reply: The larger molecular ion has the higher vibrational-state density. The vibrational autodetachment (VAD) rate constant is determined with

$$\frac{\mu}{\pi^2 \hbar^3} \int_0^{E_v - EA(J)} \frac{\sigma(\epsilon) \epsilon \rho_0(E_v^0(\epsilon))}{\rho_-(E_v)} d\epsilon$$

*where E_v, μ and ρ₋(E_v) denote the energy stored in the vibrational degrees of freedom, the reduced mass of ions and electron, and the vibrational level density of the ions, respectively; ρ₀(E_v⁰) is the corresponding vibrational level density (equation 2 in Menk, et al., *PRA* 2014, 89, 022502, cited as ref. 38). E_v⁰(ε) = E_v - EA(J) - ε, representing the vibrational energy remaining in the VAD decay product. Therefore, a higher vibrational-state density can lead to the slower autodetachment rate of anions. Above discussion is supplemented in the revised manuscript.*

(c) What electronic structure package was used to produce the potential energy curves in Fig. 3? MOLPRO? In fact, I think the authors should give more details of the calculations in the Supplemental material, enough so that Fig. 3 could be reproduced, which it could not be with the level of detail currently given.

*Reply: Our calculations were performed with a modified version of MOLPRO. We used the method as employed in *J. Mol. Struct. (Theochem)* 584, 69-77 (2002), thus the potential energy curves obtained in this work (Fig. 3) are nearly identical to those reported.*

REVIEWERS' COMMENTS:

Reviewer #1 (Remarks to the Author):

The authors have addressed all my concerns and the paper can be published. The grammar is still far from ideal but that can be solved by the editorial office.

Reviewer #2 (Remarks to the Author):

I find that the authors addressed in a satisfactory manner all the issues raised by the reviewers.

So, I recommend that this revised manuscript be accepted for publication.

Reviewer #3 (Remarks to the Author):

The authors have adequately addressed my previous comments with one exception. They acknowledge in the rebuttal letter that the program used to compute the potential curves is a modified version of Molpro, but they still do not cite Molpro in the SI. This is completely inappropriate. The authors need to add a citation to Molpro in the SI. Once this is done, then I recommend publication.

Reviewer #1:

The authors have addressed all my concerns and the paper can be published. The grammar is still far from ideal but that can be solved by the editorial office.

Reply: We modified the manuscript again.

Reviewer #2:

I find that the authors addressed in a satisfactory manner all the issues raised by the reviewers. So, I recommend that this revised manuscript be accepted for publication.

Reply: Thank you for your positive comments.

Reviewer #3 (Remarks to the Author):

The authors have adequately addressed my previous comments with one exception. They acknowledge in the rebuttal letter that the program used to compute the potential curves is a modified version of Molpro, but they still do not cite Molpro in the SI. This is completely inappropriate. The authors need to add a citation to Molpro in the SI. Once this is done, then I recommend publication.

Reply: The reference to MOLPRO, together with the others (about the theoretical method and basis set used in this work), has been added in the SI.